# Scaling up the Banded Matrix Factorization Mechanism for Differentially Private ML

**Ryan McKenna**[*]

## Abstract

Correlated noise mechanisms such as DP Matrix Factorization (DP-MF) have proven to be effective alternatives to DP-SGD in large-epsilon few-epoch training regimes. Significant work has been done to find the best correlated noise strategies, and the current state-of-the-art approach is DP-BandMF, which optimally balances the benefits of privacy amplification and noise correlation. Despite it's utility advantages, severe scalability limitations prevent this mechanism from handling large-scale training scenarios where the number of training iterations may exceed $10^4$ and the number of model parameters may exceed $10^7$. In this work, we present techniques to scale up DP-BandMF along these two dimensions, significantly extending it's reach and enabling it to handle settings with virtually any number of model parameters and training iterations, with negligible utility degradation.

## 1 Introduction

Modern machine learning is done at unprecedented scales; state-of-the-art large language models have billions of parameters and are trained on super computers with thousands of accelerators for hundreds of thousands of iterations (Kaplan et al., 2020; Chowdhery et al., 2023). Many applications would benefit from training these large-scale models on user data, but this raises a host of privacy concerns (Yao et al., 2024). Differential privacy (DP) offers a formal definition and algorithmic framework to train such models while protecting individual-level information. The most widely used DP mechanism for machine learning is DP-SGD (Abadi et al., 2016), of which there are many variants. DP-SGD relies crucially on *privacy amplification by subsampling*, and generally benefits significantly from training with huge batch sizes for many epochs (Ponomareva et al., 2023), which can be difficult or infeasible in large-scale compute-constrained settings. A promising alternative, the DP-FTRL algorithm (Kairouz et al., 2021b) instead relies on *carefully correlated noise*, and tends to work better than DP-SGD when training with smaller batches and fewer epochs, especially in the low privacy regime ($\epsilon \approx 10$) that is common in real-world scenarios.

There are many variants of the DP-FTRL algorithm that primarily differ in how they generate correlated noise (Kairouz et al., 2021b; Denisov et al., 2022; Choquette-Choo et al., 2023b; 2024; McMahan et al., 2024a; Fichtenberger et al., 2023; Henzinger et al., 2023; Kalinin & Lampert, 2024; Choquette-Choo et al., 2023a). The best variants of the DP-FTRL mechanism utilize numerical optimization to find the best noise correlation strategy, and are usually referred to as matrix factorization mechanisms (DP-MF). The space of DP-MF mechanisms is rapidly evolving, with recent work focusing intensely on improving the constant factors (Denisov et al., 2022; Choquette-Choo et al., 2023b; 2024). The variant with the best constant factors in this space is the Banded Matrix Factorization mechanism (DP-BandMF), which utilizes banded correlation matrices and enjoys the benefits of both privacy amplification and noise correlation. By setting the number of bands to one, DP-BandMF reduces to DP-SGD, and by setting the number of bands equal to the number of training iterations, DP-BandMF reduces to un-amplified variants of DP-MF. By optimally choosing the number of bands, DP-BandMF enjoys significantly better constant factors than both DP-SGD and other DP-MF-style mechanisms in many settings.

Despite its advantages, severe scalability limitations of DP-BandMF (and other DP-MF-style mechanisms more broadly) has hindered its wider use, and prevented it from scaling effectively to large-scale training regimes where typical models have billions of parameters and are trained for tens

---

[*]Google Research, `mckennar@google.com`

to hundreds of thousands of iterations. Existing DP-BANDMF implementations face computational bottlenecks stemming from two places: (1) an expensive optimization problem linked to the $n \times n$ *strategy* matrix $\mathbf{C}$ (where $n$ is the number of training iterations), requiring $O(n^3)$ time and $O(n^2)$ memory to evaluate the expected error, limiting its use to approximately $10^4$ training iterations. (2) During training, DP-BANDMF incurs an $O(b \cdot d)$ memory overhead (where $b$ is the number of bands in $\mathbf{C}$, and $d$ is model dimensionality), which is $b\times$ more than DP-SGD but $n/b\times$ less than DENSE-DP-MF. In this work, we develop techniques to overcome DP-BANDMF's scalability limitations, without sacrificing on the constant factors that make it an appealing mechanism:

- **Efficient Strategy Optimization**: We exploit the structure of banded strategies to reduce the per-iteration complexity of strategy optimization to $O(n^2 \cdot b)$ time and $O(n \cdot b)$ space, which allows DP-BANDMF to scale to approximately $n = 10^5$ training iterations. For scenarios demanding even more iterations (up to and beyond $n > 10^7$), we restrict attention to banded Toeplitz strategies, whose structure enables $O(n \cdot b)$ time and $O(n)$ space complexity during strategy optimization. Experiments show the approximation quality loss to be less than $2\%$ in terms of expected error.

- **Distributed Noise Generation**: Prior implementations of DP-MF-style mechanisms add noise on a single machine, even when training is coordinated across 1000's of machines (Denisov et al., 2022; Choquette-Choo et al., 2023b). We show how to effectively distribute the noise generation process, allowing DP-BANDMF to effectively take advantage of multi-machine environments common in large scale training regimes, and scale to larger models and more bands than was previously possible. Experiments demonstrate negligible training-time overhead compared to DP-SGD, even with hundreds of bands.

- **State-of-the-art Performance:** We conduct comprehensive experiments on expected error, and show that our scalable DP-BANDMF mechanism offers lower expected error than all other scalable MF-style mechanisms across all settings tested, including DP-SGD, and concurrent works that also improve scalability of DP-MF (McMahan et al., 2024a;b; Kalinin & Lampert, 2024).

## 2 BACKGROUND

We assume the reader has familiarity with differential privacy (Dwork, 2006; 2008). Below we provide background on DP-BANDMF, which includes DP-SGD and DP-MF-style mechanisms as special cases. DP-BANDMF is completely characterized by a $b$-banded lower triangular *strategy matrix*[1] $\mathbf{C} \in \mathbb{R}^{n \times n}$ (i.e., $C_{ij} = 0$ if $i < j$ or $i \leq j + b$).

### 2.1 TRAINING DYNAMICS

DP-BANDMF (Algorithm 2) is similar to the more well known DP-SGD algorithm (Abadi et al., 2016): minibatches are sampled randomly, gradients are clipped and aggregated, and noise is added to preserve privacy. The key innovation of DP-BANDMF is in the noise addition step: while DP-SGD adds i.i.d. Gaussian noise in each iteration, DP-BANDMF adds noise from a richer class. Specifically, the strategy matrix $\mathbf{C}$ is used to generate noise that is correlated across iterations, and the key subroutine for generating this correlated noise is shown in Algorithm 1.

When instantiated with $\mathbf{C} = \mathbf{I}$ (the identity matrix, $b = 1$), Algorithm 2 is identical to DP-SGD, as it is easy to verify the output of Algorithm 1 is simply the i.i.d. Gaussian noise vectors $\mathbf{z}_i$ passed as input. When instantiated with a dense lower triangular strategy matrix $\mathbf{C}$ ($b = n$), Algorithm 2 captures

---

[1]sometimes simply referred to as the "strategy".

---

**Algorithm 1** Banded Inverse Multiplication (Choquette-Choo et al., 2024)

**Input:** lower triangular $b$-Banded matrix $\mathbf{C} \in \mathbb{R}^{n \times n}$, stream of vectors $\mathbf{z}_1, \ldots, \mathbf{z}_n \in \mathbb{R}^d$, where $\mathbf{Z}$ is the matrix with rows $\mathbf{z}_i$.
**Output:** $\mathbf{Y} = \mathbf{C}^{-1}\mathbf{Z}$, one row at a time.
**for** $i = 1, \ldots, n$ **do**
    $\mathbf{y}_i = (\mathbf{z}_i - \sum_{j=i-b+1}^{i-1} C_{ij}\mathbf{y}_j)/C_{ii}$
    **yield** $\mathbf{y}_i$

---

**Algorithm 2** Iterative Machine Learning with DP-BANDMF

---

**Input:** $b$-banded strategy $\mathbf{C} \in \mathbb{R}^{n \times n}$, noise multiplier $\sigma$, loss function $\ell$, initial parameters $\boldsymbol{\omega}_0$, dataset $D$, expected participations $k$, clipping norm $R$
**Output:** Learned parameters $\boldsymbol{\omega}_n$
Partition $D$ into $b$ (approximately) equal sized subsets $D_0, \ldots D_{b-1}$ arbitrarily
**for** $\mathbf{y}_i$ in Algorithm 1($\mathbf{C}, \mathbf{z}_i \sim_{i.i.d} \mathcal{N}(0,1)^d$) **do** ▷ Stream of correlated noise
  Sample $B \subseteq D_{i \mod b}$ by Poisson sampling each example with probability $bk/n$.
  $\mathbf{g}_i = \sum_{x \in B} \text{clip}(\nabla \ell(\boldsymbol{\omega}_{i-1}; x), R)$ ▷ Clipped + aggregated gradient
  $\tilde{\mathbf{g}}_i = \mathbf{g}_i + R\sigma \mathbf{y}_i$ ▷ Noisy (privatized) gradient
  $\boldsymbol{\omega}_i = \text{Update}(\boldsymbol{\omega}_{i-1}, \tilde{\mathbf{g}}_i)$ ▷ Post-processing noisy gradient using any optimizer
**return** $\boldsymbol{\omega}_n$

---

DENSE-DP-MF (Denisov et al., 2022). DP-SGD benefits from privacy amplification by sampling but uses uncorrelated noise while DENSE-DP-MF benefits from correlated noise, but does not enjoy any privacy amplification due to sampling. DP-SGD tends to work better in the small-epsilon, many-epoch regime, while DENSE-DP-MF tends to work better in the large-epsilon, few-epoch regime (Choquette-Choo et al., 2024). By using a $b$-banded strategy matrix, DP-BANDMF operates between these two extremes and benefits from *both* privacy amplification by sampling and correlated noise. Specifically, the proposition below states that the privacy properties of DP-BANDMF relate very naturally to the simpler DP-SGD mechanism whose privacy properties are well-understood.

**Proposition 2.1** (Noise Calibration (Choquette-Choo et al., 2024))**.** *Let* $\sigma_{SGD}(\epsilon, \delta, k, n)$ *denote the noise multiplier required for* DP-SGD *to achieve* $(\epsilon, \delta)$-*DP when run for* $n$ *iterations with sampling probability* $k/n$. *Given a* $b$-*banded strategy matrix* $\mathbf{C}$ *satisfying* $\|\mathbf{C}\|_{1,2} \leq 1$, *Algorithm 2 satisfies* $(\epsilon, \delta)$-*DP for* $\sigma_{BandMF} = \sigma_{SGD}(\epsilon, \delta, k, n/b)$.

The function $\sigma_{SGD}$ is typically computed using numerical privacy accounting methods such as the PLD accountant (Sommer et al., 2018; Doroshenko et al., 2022).

**Remark 2.1** (Memory Overhead)**.** *Algorithm 1 requires storing a state of size* $b \times d$ ($\mathbf{y}_{i-1}, \ldots, \mathbf{y}_{i-b+1}$), *where* $b$ *is the number of bands and* $d$ *is the model dimensionality. This is* $b\times$ *larger than* DP-SGD, *but* $n/b\times$ *smaller than* DENSE-DP-MF. *For large models* ($d \geq 10^8$) *and many bands* ($b \geq 100$), *this can require hundreds of Gigabytes of space.*

## 2.2 STRATEGY OPTIMIZATION

In this section, we describe how DP-BANDMF optimally selects the number of bands $b$ and the strategy matrix $\mathbf{C}$. The matrix $\mathbf{C}$ and it's number of bands is chosen by solving a numerical optimization problem to minimize the expected total squared error added to the (clipped + aggregated) minibatch gradients during training.

**Proposition 2.2** (Expected Error (Choquette-Choo et al., 2024))**.** *The expected total squared error of* DP-BANDMF *given a* $b$-*banded strategy matrix* $\mathbf{C}$ *is equal to:*

$$\mathbb{E}[\|\mathbf{AC}^{-1}\mathbf{Z}\|_F^2] = \sigma_{BandMF}^2(\epsilon, \delta, b, k, n)\|\mathbf{C}\|_{1,2}^2\|\mathbf{AC}^{-1}\|_F^2$$

*where* $\|\mathbf{C}\|_{1,2}$ *is the maximum* $L_2$ *column norm of* $\mathbf{C}$ *and is related to it's sensitivity, and* $\mathbf{A}$ *is the workload, typically taken to be the lower triangular matrix of ones (Kairouz et al., 2021b), and* $\sigma_{BandMF}(\epsilon, \delta, b, k, n) = \sigma_{SGD}(\epsilon, \delta, k, n/b)$ *is the noise multiplier required to run* DP-BANDMF *under the given privacy budget.*

The optimization problem at the heart of DP-BANDMF follows immediately from this expression and is stated below. Note that smaller values of $b$ give better privacy amplification (smaller $\sigma_{BandMF}$), but reduces the benefits of correlated noise (more restricted space of strategies). Since $b$ is a discrete parameter, we solve the un-amplified version of the problem for fixed $b \subseteq [1, n]$ (not accounting for $\sigma_{BandMF}$), and choose the strategy matrix and number of bands that minimizes expected error accounting for $\sigma_{BandMF}$ in a brute force manner.

**Problem 2.1** (DP-BANDMF Strategy Optimization (Choquette-Choo et al., 2024))**.** *Given a work-load matrix* $\mathbf{A}$, *and the desired number of bands* $b$, *the optimization problem underlying* DP-BANDMF

---

**Algorithm 3** (Efficient) Expected Total Squared Error

---

**Input:** $\boldsymbol{\Theta} \in \mathbb{R}^{b \times n}$
**Output:** $\|\mathbf{AC}(\boldsymbol{\Theta})^{-1}\|_F^2$
$\mathbf{b}_0 = \mathbf{0}$
$\text{loss} = 0$
**for** $i = 1, \ldots, n$ **do**
  $\mathbf{e}_i = i^{th}$ row of the $n \times n$ identity matrix
  $c_i = \sqrt{\sum_{j=1}^b \Theta_{ji}^2}$        $\triangleright$ Column normalization constant
  $\mathbf{y}_i = c_i(\mathbf{e}_i - \sum_{j=1}^b \Theta_{ji}\mathbf{y}_j)/\Theta_{1i}$      $\triangleright i^{th}$ row of $\mathbf{C}(\boldsymbol{\Theta})^{-1}$
  $\mathbf{b}_i = \mathbf{b}_{i-1} + \mathbf{y}_i$         $\triangleright i^{th}$ row of $\mathbf{AC}(\boldsymbol{\Theta})^{-1}$
  $\text{loss} += \mathbf{b}_i^\top \mathbf{b}_i$        $\triangleright$ partial squared Frobenius norm
**return** loss

---

*can be formulated as:*

$$\underset{\mathbf{C}}{minimize} \quad \|\mathbf{C}\|_{1,2}^2 \|\mathbf{AC}^{-1}\|_F^2 \quad subject\ to\ \mathbf{C}_{ij} = 0 \ \ if\ \ (j > i)\ or\ (i \le j + b) \tag{1}$$

Absent the bandedness constraint, this optimization problem has received considerable attention (Li et al., 2015; Yuan et al., 2016; McKenna et al., 2023; Denisov et al., 2022), and the algorithm for solving it in the banded case is closely related (Choquette-Choo et al., 2024). Prior work (Choquette-Choo et al., 2024) solves Problem 2.1 by reformulating it in terms of the object $\mathbf{X} = \mathbf{C}^\top \mathbf{C}$. This reformulation is convex and unconstrained, meaning standard optimization tools like L-BFGS can efficiently find the best solution with access to an oracle that computes the loss and gradient with respect to $\mathbf{X}$. We note that the problem as it is stated in Problem 2.1 is *not* convex with respect to $\mathbf{C}$ (Li et al., 2015; Yuan et al., 2016; McKenna et al., 2023; 2018). The drawback of this solution approach is that it materializes several $n \times n$ matrices in each iteration, even though there are only roughly $b \cdot n$ free (non-zero) variables. As a result, it requires $O(n^3)$ time and $O(n^2)$ space to evaluate the objective once, and cannot be practically done beyond $n \approx 10^4$.

## 3   SCALABLE STRATEGY OPTIMIZATION AND NOISE GENERATION

In this section, we propose two approaches to *implicitly* compute the objective function, bypassing the need to materialize $n \times n$ matrices explicitly. These innovations enable scalability up to $n \approx 10^5$ and $n > 10^7$ respectively. For simplicity, we specialize the presentation to the Prefix workload (lower triangular matrix of ones), banded strategies, and the expected total squared error objective (Proposition 2.2). In App. D, we show that the same core approach taken here easily translates to more general classes of workloads, strategies, and objective functions.

### 3.1   EFFICIENT STRATEGY OPTIMIZATION

The complexity of solving Problem 2.1 using L-BFGS is proportional to the complexity of evaluating the objective function and its corresponding gradient. Our first key idea to scale up DP-BANDMF is to utilize Algorithm 1 to efficiently evaluate the objective function. Specifically, we will materialize one row of $\mathbf{AC}^{-1}$ at a time and compute the norm in an online fashion as shown in Algorithm 3.

Let $\boldsymbol{\Theta}$ be a $b \times n$ matrix of parameters which will define our banded strategy matrix $\mathbf{C}(\boldsymbol{\Theta})$. We define $\mathbf{C}(\boldsymbol{\Theta})$ to be the column-normalized $b$-banded matrix satisfying: $C_{ij} = \Theta_{(j-i+1)j}/\sqrt{\sum_{j=1}^b \Theta_{ji}^2}$. By construction, we have $\|\mathbf{C}(\boldsymbol{\Theta})\|_{1,2} = 1$. Thus, to evaluate the objective function we simply need to compute $\|\mathbf{AC}(\boldsymbol{\Theta})^{-1}\|_F^2$, and Algorithm 3 gives an algorithm for doing that efficiently. The key idea is to generate rows of $\mathbf{AC}(\boldsymbol{\Theta})^{-1}$ one at a time and compute the Frobenius norm in a streaming manner.

Each iteration of Algorithm 3 requires $O(n \cdot b)$ time, as the vectors $\mathbf{e}_i$ and $\mathbf{y}_i$ are size $n$, and there is a sum over $b$ such vectors. Thus, the total time complexity of Algorithm 3 is $O(n^2 \cdot b)$. The algorithm

must keep track of $\mathbf{y}_{i-b}, \ldots, \mathbf{y}_i$ in each iteration, and hence the space complexity is $O(n \cdot b)$. This is a factor $\frac{n}{b}$ improvement in both the time and space complexity of DP-BANDMF, which can be substantial when the number of bands is small (which it often is, as we demonstrate in Section 4). While Algorithm 3 is specialized to the class of (column-normalized) banded strategies and the Prefix workload, the core approach is easy to generalize to other parameterized classes of strategies and more general workloads, as we discuss in App. D.

Our L-BFGS algorithm also requires the gradient of the objective function to run efficiently. We do not derive that here, but rather rely on Jax's reverse-mode auto-differentiation capabilities instead (Bradbury et al., 2018). Depending on how the gradients are computed, the gradient computation can be more time and memory intensive than the loss calculation. We discuss this more in depth in App. C and provide some experiments on the scalability of this approach in Section 4. Using these ideas, we can scale DP-BANDMF up to roughly $n \approx 10^5$ under reasonable time and memory constraints, without sacrificing any solution quality over prior work (Choquette-Choo et al., 2024).

## 3.2 OPTIMIZING BANDED TOEPLITZ STRATEGIES

We now explore another technique that is even more scalable, while only sacrificing a small amount in terms of expected error. Our key idea is to restrict attention to the special class of *banded Toeplitz strategies*. This design decision was inspired by manual inspection of the optimal dense strategies, observing that they exhibit a near-Toeplitz structure. By exploiting the special structure of this class of strategies, we can evaluate the objective function using only $O(n \cdot b)$ time and $O(n)$ space, and scale strategy optimization up to and beyond $n \approx 10^6$, as we show below:

**Proposition 3.1** (Banded Toeplitz Expected Total Squared Error). *Let $\boldsymbol{\theta} \in \mathbb{R}^b$ be Toeplitz coefficients, such that $\mathbf{C}(\boldsymbol{\theta})_{ij} = \theta_{i-j+1}$, and let $\mathbf{w} = \mathbf{C}(\boldsymbol{\theta})^{-1}\mathbf{1}$. The expected total squared error is equal to:*

$$\|\mathbf{C}(\boldsymbol{\theta})\|_{1,2} = \|\boldsymbol{\theta}\|_2 \qquad \|\mathbf{A}\mathbf{C}(\boldsymbol{\theta})^{-1}\|_F^2 = \sum_{i=1}^{n}(n-i+1)w_i^2$$

*Proof.* The first expression states that the maximum $L_2$ column norm of a lower triangular Toeplitz matrix is equal to the norm of it's first column $\boldsymbol{\theta}$. By direct inspection, the first $n - b + 1$ columns of $\mathbf{C}(\boldsymbol{\theta})$ all share the same non-zero entries of $\boldsymbol{\theta}$, and the norms of these columns are thus $\|\boldsymbol{\theta}\|_2$. The non-zero entries of the remaining $b - 1$ columns of $\mathbf{C}(\boldsymbol{\theta})$ correspond to a subvector of $\boldsymbol{\theta}$, and hence these columns have norm $\leq \|\boldsymbol{\theta}\|_2$. The second expression states that the Frobenius norm can be calculated without explicitly materializing any matrices. Note that both $\mathbf{A}$ and $\mathbf{C}(\boldsymbol{\theta})$ are lower triangular Toeplitz matrices, and if $\mathbf{C}(\boldsymbol{\theta})$ is invertible, then its inverse is also a lower triangular Toeplitz matrix (Lin, 2008, Lemma 5). Second note that multiplication is commutative within this class, and therefore $\mathbf{A}\mathbf{C}^{-1}(\boldsymbol{\theta}) = \mathbf{C}^{-1}(\boldsymbol{\theta})\mathbf{A}$, and this product is itself a lower triangular Toeplitz matrix (Lin, 2008, Lemma 5). As such, it is completely defined by its first column $\mathbf{w}$. Using the definition of matrix multiplication we can obtain $\mathbf{w}$ by multiplying $\mathbf{C}^{-1}(\boldsymbol{\theta})$ by the first column of $\mathbf{A}$, which is $\mathbf{1}$ (Lay, 2003, Page 97). The squared Frobenius norm of a Toeplitz matrix with parameters $\mathbf{w}$ can be calculated directly over this vector by observing that $w_i$ is repeated along the $i^{th}$ diagonal band of the Toeplitz matrix and hence appears $(n - i + 1)$ times, leading to the expression stated above for the squared Frobenius norm of $\mathbf{A}\mathbf{C}^{-1}(\boldsymbol{\theta})$. $\square$

We note that this approach generalizes to any Toeplitz-structured workload, as we show in App. D. The complexity of evaluating the objective function is determined by the cost of solving the linear Toeplitz system $\mathbf{w} = \mathbf{C}(\boldsymbol{\theta})^{-1}\mathbf{1}$, which can be done with Algorithm 1 in $O(n \cdot b)$ time and $O(b)$ space. In App. I we further reduce the time complexity to $O(b^3)$ by observing that the sequence $w_i$ rapidly converges to a fixed point, which allows scalability to virtually any $n$ (as long as $b$ is not too large).

**Column Normalization** In general the column norms of Toeplitz matrices are not all equal, a property that optimal strategies are known to have in the single-participation setting (Zhang et al., 2018; Yuan et al., 2016; McKenna et al., 2018) and a property built-in to DP-BANDMF (Choquette-Choo et al., 2024). When the number of bands is small, the loss in solution quality from not column normalizing is small, as the first $n - b$ columns of a banded Toeplitz strategy already have equal norm. Moreover, we can always normalize the columns of the banded Toeplitz matrices as a post-processing step after strategy optimization, which we recommend in practice.

### 3.3 Distributed Noise Generation

At training time, DP-BANDMF incurs a $b \cdot d$ memory overhead where $b$ is the number of bands and $d$ is the number of model parameters. In large-scale settings $d$ may exceed $10^9$, and assuming $b = 256$ and 4-byte floats are used, this is a very significant one *terabyte* cost. In this section, we will argue that with a careful implementation, this is not a major issue in typical large-scale training regimes.

We make two key observations. First, in order to train large models with differential privacy, large batch sizes are typically needed, which requires a combination of coordinated *training across many accelerators* and virtual batching (De et al., 2022; Anil et al., 2022). For example, to train a 1 billion parameter model with DP, one might use 1024 machines and $16\times$ virtual batching to attain a reasonable batch size of 16384 (assuming a per core batch size of 1). Second, we can effectively utilize multiple machines when generating noise by observing that Algorithm 1 can be translated to an *embarrassingly parallel* algorithm since each coordinate of the noise vectors are treated identically.

In our implementation, each machine is in charge of producing a different shard of the noise vectors $\mathbf{y}_i$, and they keep track of the state required to generate that noise locally.[2] This state corresponds to the same shard of the previous $b - 1$ correlated noise vectors $(\mathbf{y}_{i-1}, \ldots, \mathbf{y}_{i-b+1})$, and this sharding strategy is illustrated in the figure to the right. This benefit of this sharding strategy is that no communication is needed between machines until the noise is added to the clipped + aggregated gradient. Using the example above, this implementation would have a very manageable overhead of one GB per machine (one TB split across 1024 machines).

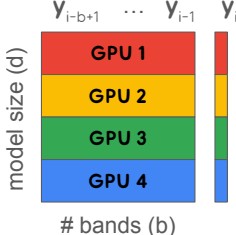

In Section 4.2 we demonstrate that in practice the number of bands needed is typically far less than 256, and that the overhead incurred by our distributed correlated noise generation algorithm is small compared to the cost of per-example gradient clipping, even when the number of bands is large. To better understand the scalability limits of this distributed noise generation approach in different settings, it is useful to look at a few more illustrative examples.

**Example 3.1.** *Suppose we want to train a 4 million parameter model on a single GPU with a memory limit of 1 GB. We can afford to store roughly 64 copies of the model assuming each parameter is represented as a 4-byte float. This allows us to use roughly 64 bands overall.*

**Example 3.2.** *Now suppose we want to train a larger 128 million parameter model in parallel on 64 machines, which is useful/needed to get the large batch sizes usually used with large models and DP training. By parallelizing the noise generation code across these 64 machines, we can afford to store 2 copies of the model per machine, allowing us to use roughly 128 bands overall.*

**Example 3.3.** *Now suppose we want to finetune an 8 billion parameter model on 64 machines, and use a parameter-efficient fine-tuning method such as LoRA (Hu et al., 2021; Yu et al., 2021). Further, assume the number of learnable finetuning parameters is 4 million. Now, the cost of the forward/backward pass completely dwarfs any overhead of noise generation. We can store 64 copies of the model per machine, and parallelize computation across 64 machines, allowing us to run DP-BANDMF with up to 4096 bands.*

## 4 Empirical Results

In this section, we empirically evaluate our proposed strategy optimization algorithms in terms of solution quality and scalability. Then, we conduct experiments and perform analysis to understand the optimal number of bands as a function of the relevant parameters. Our analysis reveals that in many scenarios of practical interest the optimal number of bands is small, and consequently the overhead of DP-BANDMF at training time is small. Our experiments focus on root-mean-squared-error (RMSE), which we compute in closed form using Proposition 2.2, dividing by $n$ and taking the square root. Strategy optimization is done on an NVIDIA V100 Tensor Core GPU for up to 10K iterations.

---

[2]this strategy for distributing noise generation is very different from distributed differential privacy (Kairouz et al., 2021a), where each client/machine generates an entire vector of noise with smaller variance.

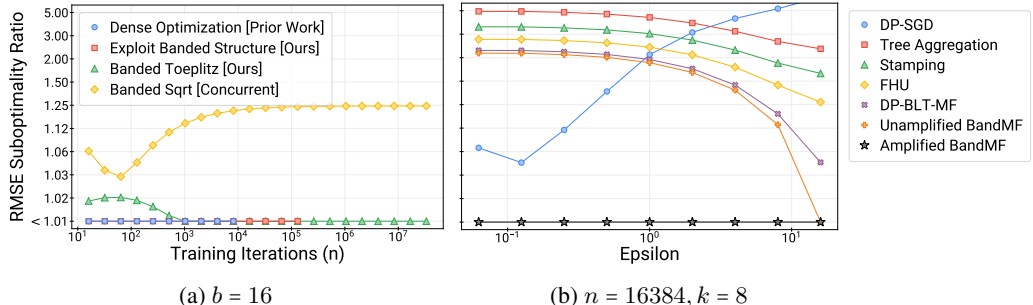

(a) $b = 16$                    (b) $n = 16384, k = 8$

Figure 1: (a-b) Ratio of RMSE of each strategy to the best strategy that scales to each setting. (a) Compares our scalable banded and banded toeplitz strategies with other banded strategies. (b) Compares our scalable DP-BANDMF with non-banded strategies as a function of $\epsilon$.

## 4.1 COMPARISON TO PRIOR AND CONCURRENT WORK

We begin by evaluating the solution quality of our strategies compared to prior and concurrent work. We provide a thorough qualitative comparison between the mechanisms considered here in App. G. In Figure 1a, we fix $b = 16$ while varying $n$, and compare our strategies obtained through implicit optimization to the strategies produced through explicit optimization as in prior work (Choquette-Choo et al., 2024). We also include BANDED SQRT as an additional baseline, which was proposed in concurrent work (Kalinin & Lampert, 2024), and avoids dense matrix representations through the same banded Toeplitz structure we consider. We plot on the y-axis the *RMSE Suboptimality Ratio*, which we define as the ratio of RMSE between a given strategy and the best available strategy for that setting. Because all the strategies in this figure are 16-banded, they enjoy the same privacy properties (whether they are used in an amplified or non-amplified setting), and therefore the RMSE Suboptimality Ratio's we report hold for all $\epsilon$. The highlights from this experiment are four-fold:

- The prior work that represents the strategy using dense matrices scaled up to $n < 10^4$, while our implicitly optimized banded strategies scaled up to $n > 10^5$, and our banded Toeplitz strategies scaled well beyond $n > 10^7$. We provide more detailed scalability results in Figure 4 of App. F.

- Our implicit strategy optimization algorithm for general banded matrices converges to the same solution as prior work (Choquette-Choo et al., 2024) for all settings tested (where the latter successfully ran). This is encouraging in light of the fact that we directly optimize $\mathbf{C}$ (rather than $\mathbf{X} = \mathbf{C}^\top\mathbf{C}$) and Problem 2.1 is not convex with respect to $\mathbf{C}$.

- The BANDED SQRT baseline approach (Kalinin & Lampert, 2024) achieves worse expected error across all settings than both of our approaches. The suboptimality approaches up to $25\%$ in the settings we tested, with larger suboptimality for larger $n$.

- Our (column-normalized) Toeplitz strategies are between $0$-$2\%$ suboptimal, with suboptimality increasing with the number of bands and decreasing with the number of iterations. For large $n \geq 16384$ and small $b \leq 32$, which is the regime of most interest, the suboptimality is $\leq 0.25\%$, indicating arbitrary banded strategies do not provide much benefit over the more restrictive class.

In Figure 1b we compare against several additional baseline mechanisms for $n = 16384$ iterations and $k = 8$ epochs while varying $\epsilon$. We include results for DP-SGD (Abadi et al., 2016), Tree Aggregation (Kairouz et al., 2021b), Stamping (Denisov et al., 2022), FHU (Fichtenberger et al., 2023), Buffered Toeplitz, (McMahan et al., 2024a;b), and our Unamplified and Amplified scalable DP-BANDMF approach. We omit from comparison DENSE-DP-MF (Choquette-Choo et al., 2023b) due to scalability limitations of that approach for $n \geq 16384$.

Here, DP-SGD and Amplified DP-BANDMF benefit from privacy amplification, while the other mechanisms do not. Our highlights from this experiment are two-fold:

- Amplified DP-BANDMF is better than all other mechanism across all settings evaluated. The improvement over DP-SGD grows with $\epsilon$, while the improvement over other mechanisms decreases with $\epsilon$. The best alternatives are either DP-SGD or Buffered Toeplitz, both of which have $\approx 2\times$

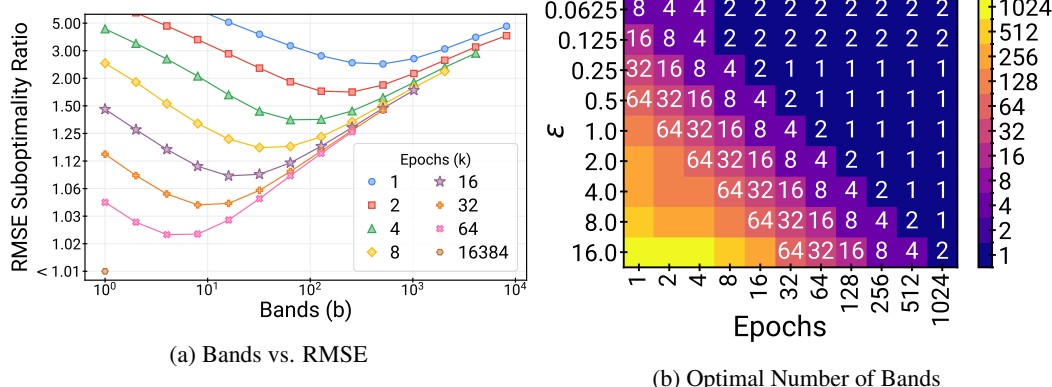

(a) Bands vs. RMSE

(b) Optimal Number of Bands

Figure 2: (a) RMSE Suboptimality Ratio (relative to full-batch DP-SGD) of DP-BANDMF as a function of $b$ for various epochs, with fixed $(\epsilon, \delta) = (1, 10^{-8})$ and $n = 16384$. (b) Optimal number of bands (within a factor of 2) as a function of the privacy budget and the number of epochs, fixing $n = 4096$ and $\delta = 10^{-8}$.

worse RMSE for $\epsilon = 1$. Amplified DP-BANDMF still enjoy a meaningful 19% better RMSE than the best competitor for $\epsilon$ as large as 8.

- Unamplified DP-BANDMF is the best non-amplified mechanism that can scale to this setting, and enjoys ~ 5% lower RMSE than Buffered Toeplitz strategies for all $\epsilon$. While our primary focus has been in centralized training regimes, this figure shows that DP-BANDMF is also a state-of-the-art approach in federated scenarios where amplification is not feasible (Kairouz et al., 2021b).

While we focused on a single value of $n$ and $k$ in Figure 1b, our general findings remain unchanged for other settings, although the magnitude of the improvements do change. For completeness, results for other values of $n$ and $k$ are shown in Figure 7 in App. F.

## 4.2 OPTIMAL NUMBER OF BANDS

We now perform an ablation on DP-BANDMF to understand how to select the number of bands in practice, and how the performance of DP-BANDMF changes as a function of the number of epochs. Figure 2a plots the RMSE suboptimality ratio of DP-BANDMF relative to the full-batch DP-SGD baseline for different values for epochs and bands fixing at $\epsilon = 1$. Figure 2b plots the optimal number of bands for different values of epochs and $\epsilon$. Our findings from these experiments are three-fold:

- With only 32 epochs, DP-SGD ($b = 1$) has nearly the same RMSE as full-batch DP-SGD (16384 epochs), but the RMSE degrades rapidly with fewer epochs. Unamplified DP-BANDMF ($b = n/k$) achieves worse RMSE than DP-SGD when the number of epochs is large (and utility is the best), but it degrades more gracefully than DP-SGD, offering better utility in the few-epoch regime. Amplified DP-BANDMF ($b = b_*$) offers the best of both worlds: full-batch DP-SGD-level performance when the number of epochs is sufficiently large, and even more graceful degradation of RMSE with respect to number of epochs than Unamplified DP-BANDMF.

- The RMSE improves predictably with increasing epochs, but there are diminishing marginal returns. For example, the RMSE of Amplified DP-BANDMF is < 2× worse than the RMSE of full-batch DP-SGD with only two epochs, and < 18% worse with only eight epochs. We note that RMSE only accounts for the noise added due to privacy; increasing the number of epochs under a fixed number of iterations increases the batch size, which also reduces the minibatch gradient variance.

- The optimal number of bands drops predictably with increasing epochs and decreasing epsilon. Specifically, the dependence appears roughly linear in both parameters, and a good rule of thumb is that $b_* \approx \epsilon\sqrt{n}/k$ should be near-optimal. In some parameter regimes, like $k = 1, \epsilon \geq 16$, $b_*$ may be too large to feasibly handle. However, Amplified DP-BANDMF with a suboptimal number of bands is still better than the alternative of DP-SGD.

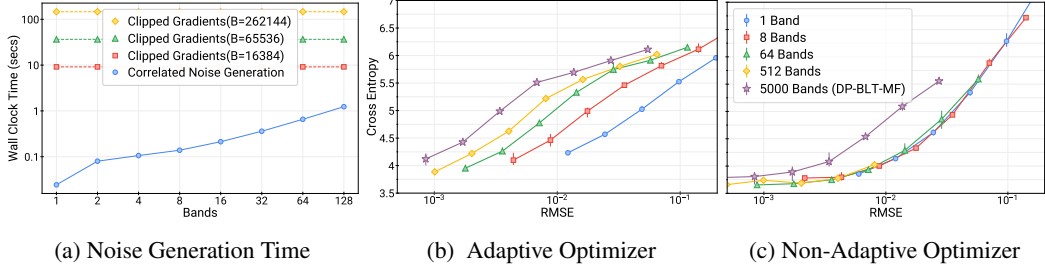

(a) Noise Generation Time    (b)  Adaptive Optimizer    (c) Non-Adaptive Optimizer

Figure 3: (a) Wall Clock Time for correlated noise generation and per-example gradient clipping for a 100M parameter `BertBase` model when run on 32 TPU v3 cores. (b-c) RMSE vs. Learning Performance (evaluation cross entropy) with an adaptive optimizer (b) and a non-adaptive optimizer (c). In both (b) and (c), a 4M parameter `BertTiny` model is trained on the StackOverflow dataset for various noise multipliers.

The analysis above reveals what the optimal number of bands is for different settings, but not whether it is feasible to run the mechanism for that many bands. In Figure 3a we plot the wall clock time to generate correlated noise as a function of the number of bands in a typical distributed training scenario with a 100M parameter `BertBase` model trained on 32 accelerators. Our primary finding for this experiment is:

- Using our distributed algorithm, correlated noise generation is not the primary bottleneck, even when the number of bands is fairly large (up to $b = 128$, corresponding to $1.6$ GB overhead per accelerator). The cost of per-example gradient clipping is one- to three- orders of magnitude more expensive, depending on the batch size.

## 4.3   RMSE VS. LEARNING PERFORMANCE

Thus far we have primarily focused on RMSE on the prefix workload, but this is just a proxy for what we really care about: training-time learning performance. In Figure 3(b-c) we aim to understand the relationship between RMSE and learning performance across a range of DP-MF strategies. Our goal is not to compare different mechanisms across a different privacy budgets, as this has been done in prior work (Choquette-Choo et al., 2024; 2023a; Denisov et al., 2022), and our mechanism is just a more scalable instantiation of the prior work.

We consider strategies with differing numbers of bands, and for each setting we optimize the strategy for the prefix workload. Note this is different from early work on matrix factorization where the strategy is optimized for a workload that encodes the optimizer parameters like momentum and learning rate cooldown (Denisov et al., 2022). Recent work on DP-BANDMF suggest it is fine to configure the strategy for the prefix workload even when using SGD with momentum in practice (Choquette-Choo et al., 2024, Section I).

For each strategy under consideration, we train a `BertTiny` model with per-example clipping and correlated noise addition with a range of noise multiplies spanning 3 orders of magnitude. We tune the learning rate for each setting and report results for the best value found. We each experiment, we record the RMSE of the strategy / noise multiplier along with the evaluation set cross entropy, and plot the results in Figures 3b and 3c. Our two main findings from this experiment are:

- For an adaptive optimizer and the same RMSE, *strategies with fewer bands provide better learning performance*, indicating that RMSE is not the best proxy for learning performance across multiple strategies. These results suggest that one should set bands more conservatively than suggested by RMSE alone to optimize learning performance if using DP-BANDMF with an adaptive optimizer.

- For a non-adaptive optimizer, the lines for 1-, 8-, 64-, and 512-banded strategies all line up, indicating that RMSE is a reasonable predictor of learning performance. However, there is still a gap between DP-BANDMF and BLT-DP-MF (5000 bands), where fewer bands achieves better learning performance for the same RMSE.

While this experiment focused RMSE, we also tested the effectiveness of max error as a proxy for learning performance in Figure 9 in App. F, with similar observations holding there as well.

## 5 RELATED WORK

**More DP-MF Variants**    Koloskova et al. (2024); Choquette-Choo et al. (2023a) revisit the objective underlying DP-MF and propose new variants based on their analysis. Like us, Henzinger et al. (2023) and Choquette-Choo et al. (2023a) use Toeplitz strategies, and bypass the $O(n^3)$ runtime of strategy optimization. However, they consider "full" Toeplitz matrices and do not address the $O(n \cdot d)$ training-time memory overhead of the algorithm, preventing scalability in large-scale settings.

**Scalable Approaches to the Matrix Mechanism**    The optimization problem we studied in this work dates back to the Matrix Mechanism (Li et al., 2010), and has been the subject of intense research for a different problem domain: answering workloads of linear counting queries over discrete databases (Li et al., 2010; 2015; Yuan et al., 2016; McKenna et al., 2018; 2023; Xiao et al., 2024; Nikolov et al., 2013; Edmonds et al., 2020; McKenna et al., 2020; Li & Miklau, 2013; Xiao et al., 2024). McKenna et al. (2023) and Xiao et al. (2024) restrict attention to special classes of strategies that enable efficient strategy optimization. By exploiting the special structure in these strategies, they scale well beyond $n > 10^7$, but the specific strategies considered in those works do not directly apply in the context of iteratively training ML models.

Concurrently with our work, McMahan et al. (2024a) proposed BLT-DP-MF, a memory-efficient approximation of single-epoch DENSE-DP-MF, and extended to handle multiple epochs in follow-up work (McMahan et al., 2024b). Their method scales to an arbitrary number of training iterations, and reduces the training-time memory complexity to $O(c \cdot d)$ for small and configurable $c \approx 3$, while only sacrificing 5-7.5% in RMSE compared to Unamplified DP-BANDMF.

## 6 LIMITATIONS

Here we discuss several limitations of this paper. First, the number of bands needed to maximize utility may differ from the number of bands that is feasible to use under compute constraints. In Section 4.2 we provided an argument that in the regimes of most practical interest, the number of bands will be small. However, it is conceivable for there to be situations when the optimal number of bands is larger than what can be supported under compute constraints.

Second DP-BANDMF relies on Poisson sampling to form minibatches, which can be tricky to integrate into modern ML infrastructure due to variable batch sizes.

Third, this work and nearly all work on DP Matrix Factorization relies on the assumption that expected error on the prefix sums of gradients is a good proxy for learning performance (Kairouz et al., 2021b; Denisov et al., 2022; Choquette-Choo et al., 2023a; McMahan et al., 2024a; Kalinin & Lampert, 2024). As demonstrated in Section 4.3, this proxy is not perfect, particularly when combined with adaptive optimizers. More research is needed to improve the objective function underlying DP-MF-style mechanisms, and how to configure them with adaptive optimizers.

Finally, our distributed noise generation procedure relies on the assumption that the worker machines responsible for computing each shard of the noise are not compromised, otherwise the privacy guarantees could be at risk. In situations where noise must be generated on a single machine, DP-BANDMF may not be able to scale to as many bands. However, as we showed throughout Section 4, DP-BANDMF tends to work best with a small number of bands anyway.

## 7 CONCLUSION

This work is motivated by the desire to train large-scale ML models with differential privacy. Our work is best characterized by its simplicity, scalability, and state-of-the-art performance. Our key ideas are straightforward to understand and implement. Our algorithms for strategy optimization scale effectively well beyond $n > 10^7$, and our algorithm for distributed noise generation can handle large models with little overhead. Finally, our mechanism offers better expected error than any other DP-MF-style mechanism across a wide variety of settings.

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

## A  NOTATION

| Symbol | Domain | Meaning |
|--------|--------|---------|
| $n$ | $\mathbb{N}_+$ | number of training iterations |
| $d$ | $\mathbb{N}_+$ | model dimensionality |
| $k$ | $\mathbb{N}_+$ | number of epochs |
| $q$ | $(0,1]$ | minibatch sampling probability |
| $b$ | $\mathbb{N}_+$ | number of bands |
| $\sigma$ | $\mathbb{R}_+$ | noise multiplier |
| $\mathbf{A}$ | $\mathbb{R}^{n \times n}$ | lower triangular ones matrix |
| $\mathbf{Z}$ | $\mathbb{R}^{n \times d}$ | Gaussian noise, $Z_{ij} \sim \mathcal{N}(0, \sigma^2)$ |
| $\mathbf{C}$ | $\mathbb{R}^{n \times n}$ | DP-BANDMF strategy matrix |
| $\mathbf{C}(\boldsymbol{\theta})$ | $\mathbb{R}^{n \times n}$ | Parameterized strategy matrix |
| $\mathbf{e}_i$ | $\mathbb{R}^n$ | $i^{th}$ indicator vector |
| $\|\cdot\|_{1,2}$ | $\mathbb{R}_+$ | maximum $L_2$ column norm |
| $\sigma_{SGD}(\epsilon, \delta, q, n)$ | $\mathbb{R}_+$ | DP-SGD noise multiplier |

Table 1: Table of Notation

## B  EXAMPLES

### B.1  OPTIMAL 3-BANDED STRATEGY

**Example B.1** (DP-BANDMF)**.** *The matrix below is the optimal 3-banded strategy matrix* $\mathbf{C}$ *configured for 9 training iterations. (Informal [3]) DP-BANDMF with minibatch sampling probability of*

---

[3]DP-BANDMF requires a special sampling different from the usual Poisson sampling of DP-SGD to achieve this guarantee, see Choquette-Choo et al. (2024) for the details.

*$q \leq 1/3$ enjoys the same privacy guarantees as 3 iterations of* DP-SGD *with sampling probability* $3q$. *The vectors* $\mathbf{y}_1, \ldots \mathbf{y}_9$ *are added to the minibatch gradients during training, and can be computed efficiently in an online fashion with* $O(b \cdot d)$ *space using Algorithm 1, exemplified below:*

$$
\begin{bmatrix}
0.740 & 0 & 0 & 0 & 0 & 0 & 0 & 0 & 0 \\
0.500 & 0.822 & 0 & 0 & 0 & 0 & 0 & 0 & 0 \\
0.450 & 0.492 & 0.876 & 0 & 0 & 0 & 0 & 0 & 0 \\
0 & 0.286 & 0.395 & 0.821 & 0 & 0 & 0 & 0 & 0 \\
0 & 0 & 0.278 & 0.462 & 0.855 & 0 & 0 & 0 & 0 \\
0 & 0 & 0 & 0.335 & 0.442 & 0.882 & 0 & 0 & 0 \\
0 & 0 & 0 & 0 & 0.272 & 0.403 & 0.892 & 0 & 0 \\
0 & 0 & 0 & 0 & 0 & 0.243 & 0.409 & 0.936 & 0 \\
0 & 0 & 0 & 0 & 0 & 0 & 0.194 & 0.353 & 1.000
\end{bmatrix}^{-1}
\begin{bmatrix}
\mathbf{z}_1 \\ \mathbf{z}_2 \\ \mathbf{z}_3 \\ \mathbf{z}_4 \\ \mathbf{z}_5 \\ \mathbf{z}_6 \\ \mathbf{z}_7 \\ \mathbf{z}_8 \\ \mathbf{z}_9
\end{bmatrix}
=
\begin{bmatrix}
\mathbf{y}_1 \\ \mathbf{y}_2 \\ \mathbf{y}_3 \\ \mathbf{y}_4 \\ \mathbf{y}_5 \\ \mathbf{y}_6 \\ \mathbf{y}_7 \\ \mathbf{y}_8 \\ \mathbf{y}_9
\end{bmatrix}
$$

- $\mathbf{y}_1 = \mathbf{z}_1 / 0.740$
- $\mathbf{y}_2 = (\mathbf{z}_2 - 0.500\mathbf{y}_1) / 0.822$
- $\mathbf{y}_3 = (\mathbf{z}_3 - 0.492\mathbf{y}_2 - 0.450\mathbf{y}_1) / 0.876$
- $\mathbf{y}_4 = (\mathbf{z}_4 - 0.395\mathbf{y}_3 - 0.286\mathbf{y}_2) / 0.821$
- $\ldots$
- $\mathbf{y}_9 = (\mathbf{z}_9 - 0.353\mathbf{y}_8 - 0.194\mathbf{y}_7) / 1.000$

## C  COMPUTING THE GRADIENT OF THE EXPECTED ERROR

We implemented functions to compute the expected error of banded strategies (Algorithm 5) and banded Toeplitz strategies (Proposition 3.1) in pure JAX. This allowed us to use the `jax.grad` transformation to give us a function to compute its gradients, so we didn't have to manually derive them. In this section, we discuss the time and memory complexity of these gradient calculations.

For both banded strategies and banded Toeplitz strategies, computing the gradient of the objective function requires back-propagation through Algorithm 1. In the banded case, this algorithm is invoked with a sequence of size $n$ vectors, while in the banded Toeplitz case, it is invoked with a sequence of scalars. To compute the objective function, the intermediate values $\mathbf{y}_i$ can be discarded on step $b + i$. However, to compute the gradient, by default `jax.grad` requires keeping around all values of $\mathbf{y}_i$ in memory at once.

**Banded Strategies** Both the time and memory complexity of computing the gradient with this default implementation is $O(n^2 b)$ in the banded case. By using the `jax.checkpoint` transformation, we can trade-off memory for time during the backward pass by only storing some subset of intermediates. Specifically, given a parameter $k$ that roughly corresponds to the number of intermediates to keep around for the backwards, we can compute the gradient using $O(n^2 bk)$ time and $O(n^2 b/k)$ memory by re-materializing the inputs we need when we need them. We choose $k$ so that the total memory consumed is less than or equal to `4GB`.

**Banded Toeplitz Strategies** The time and memory complexity of computing the gradient with the default implementation is $O(bn)$ and $O(n)$ respectively. We believe the memory complexity of he loss function could be reduced to $O(b)$ with a more careful implementation though. However, $O(n)$ memory is already small enough to scale up to all values of $n$ we care about in practice, and no special tricks are needed to improve the memory footprint.

### C.1  TIMING EXPERIMENTS

Figure Figure 4 shows the wallclock time required to evaluate the loss function and it's gradient on both CPUs and GPUs, for various $n$ at fixed $b = 16$. We evaluate the wall clock time for increasing $n$ until the total time to compute both the loss and grad exceeds 60 seconds. Our findings from this experiment are:

- We can compute both the loss and gradient of general banded strategies up to about $n \approx 10^5$ under reasonable time limits when utilizing GPUs. As $n$ gets larger, the relative gap between the time to compute the loss and gradient grows, as expected due to our checkpointing strategy described above.

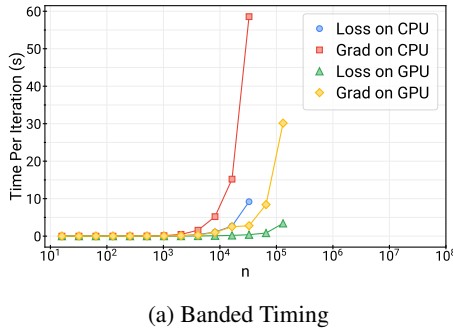 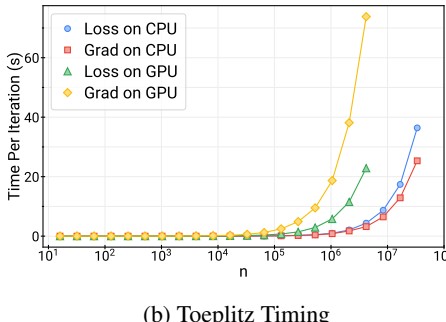

(a) Banded Timing                    (b) Toeplitz Timing

Figure 4: Wallclock time required to evaluate the total squared error objective function and it's gradient as a function of $n$ for $b = 16$. Does not include JIT-compile time, which is amortized over strategy optimization.

---

**Algorithm 4** Streaming Linear Operator for Prefix
---

> **Input:** stream of vectors $\mathbf{z}_1, \ldots, \mathbf{z}_n \in \mathbb{R}^d$, where $\mathbf{Z}$ is the matrix with rows $\mathbf{z}_i$.
> **Output:** $\mathbf{Y} = \mathbf{AZ}$, one row at a time.
> **for** $i = 1, \ldots, n$ **do**
>     $\mathbf{y}_i = \mathbf{y}_{i-1} + \mathbf{z}_i$
>     **yield** $\mathbf{y}_i$

---

Figure 5: Algorithm for streaming matrix multiplication by a Prefix matrix. To simplify the presentation, we use the convention that out-of-bounds indexing into a matrix or vector returns $0$.

- We can compute both the loss and gradient of banded Toeplitz strategies beyond $n > 10^7$ under reasonable time limits. These computations do not benefit from GPUs, and in fact an order of magnitude faster on CPUs. This is due to the inherent sequential nature of Algorithm 1, which is not a type of computation that benefits from GPUs.

## D    GENERALIZATIONS TO OTHER STRATEGY AND WORKLOAD CLASSES

Our presentation in Section 3.1 focused on the Prefix workload and the class of (column-normalized) banded matrices and Toeplitz matrices. However, our core approach also applies to a broader set of strategies in workloads, as we show in this section.

### D.1    STREAMING LINEAR OPERATORS

We begin by providing a definition for a Streaming Linear Operator:

**Definition D.1** (Streaming Linear Operator (SLO)). *A Streaming Linear Operator is a function that consumes a sequence of vectors $\mathbf{z}_1, \ldots, \mathbf{z}_n$ as input one at a time, and produces a sequence of vectors $\mathbf{y}_1, \ldots, \mathbf{y}_n$ one at a time such that $\mathbf{y}_i$ is a linear function of $\mathbf{z}_1, \ldots, \mathbf{z}_i$. We say a SLO is $b$-buffered if it only requires storing a state of size at most $b$ vectors.*

We note that there is a one-to-one correspondence between SLOs and lower triangular matrices. The Prefix matrix can be represented as a $1$-buffer SLO, shown in Figure 5. Banded lower triangular matrices and their inverses can be represented as $b$-buffer SLOs, as shown in Algorithm 1. The heavyball momentum + learning rate cooldown workload studied in prior work Denison et al. (2022); Choquette-Choo et al. (2023b) can also be represented as a 2-buffer SLO.

---

**Algorithm 5** Efficiently Evaluating the Per Query Error

---

**Input:** SLO for $C(\Theta)^{-1}$, SLO for $\mathbf{A}$
**Output:** $\mathbf{b}_i^\top \mathbf{b}_i$, one entry at a time, where $\mathbf{B} = \mathbf{A}C(\Theta)^{-1}$
$\mathbf{b}_0 = \mathbf{0}$
**for** $i = 1, \dots, n$ **do**
    $\mathbf{y}_i = SLO_{C^{-1}}(\mathbf{e}_i)$
    $\mathbf{b}_i = SLO_A(\mathbf{y}_i)$
    **yield** $\mathbf{b}_i^\top \mathbf{b}_i$

---

## D.2 EFFICIENTLY OPTIMIZING PARAMETERIZED STRATEGIES VIA SLOs

Recall that computing the objective function efficiently is the core requirement for performing efficient strategy optimization over the space of strategies. As long as we can express this function in terms of Jax primitives, we can rely on auto-differentiation tools to compute the gradients and perform the optimization. Hence, our focus in this section is on generalizing Algorithm 3 to handle *any* workload represented as a SLO, and *any* parameterized strategy whose inverse is a SLO, and *any* differentiable aggregator of the per-query expected squared errors.

Algorithm 5 is an algorithm to calculate the per-query expected error of any workload and parameterized strategy represented as a pair of SLOs. The time and memory complexity of this algorithm depends only on the complexity of the underlying SLO implementations. In general, for a $b_1$-buffer SLO for $\mathbf{C}^{-1}$ and $b_2$-buffer SLO for $\mathbf{A}$, the time and memory complexity of Algorithm 5 is $O(n^2(b_1 + b_2))$ and $O(n(b_1 + b_2))$ respectively. We note that the per-query-error can be combined with any (sub)-differentiable aggregator to get a scalar-valued loss function that can be optimized. In this paper, our focus was on expected total squared error, which is simply a sum of the per-query squared errors. However, the max expected error is another natural choice that has been sometimes used in prior work McMahan et al. (2024a); Fichtenberger et al. (2023). The "best" loss function to use, i.e., the one that best correlates with learning performance, is still an open question.

## D.3 EFFICIENTLY OPTIMIZING BANDED TOEPLITZ STRATEGIES FOR TOEPLITZ WORKLOADS

In this section, we show that we can efficiently calculate the expected per-query squared error for a banded Toeplitz strategy on *any* Toeplitz workload with a minor modification to Proposition 3.1. These generalizations do not affect the time or memory complexity of the algorithm.

**Proposition D.1** (Banded Toeplitz Expected Per-Query Squared Error). *Let $\boldsymbol{\lambda}, \boldsymbol{\theta} \in \mathbb{R}^b$ be Toeplitz coefficients for $\mathbf{A}$ and $\mathbf{C}$ respectively. The expected square error (excluding sensitivity) on the $i^{th}$ query can be evaluated as $\sum_{j=1}^i w_j^2$ where $\mathbf{w} = \mathbf{C}(\boldsymbol{\theta})^{-1}\boldsymbol{\lambda}$*

# E THEORETICAL GUARANTEES

In this section we provide theoretical guarantees on the expected error of our mechanism. To do so, it is useful to state the theoretical guarantees of the BANDED SQRT mechanism. We restate their main theoretical guarantee below, specialized to the Prefix workload:

**Proposition E.1** (Thm 6 Kalinin & Lampert (2024)). *Let $\mathbf{A}$ denote the Prefix workload let $\mathbf{C}(\boldsymbol{\theta})$ denote the strategy matrix for the* Banded Square Root *factorization with $b$ bands. Then*

$$\|\mathbf{C}(\boldsymbol{\theta})\|_{1,2}\|\mathbf{A}\mathbf{C}(\boldsymbol{\theta})^{-1}\|_F = O\left(\sqrt{\frac{n \log b}{b}}\right)$$

We will now argue that our Banded Toeplitz strategy inherits the theoretical guarantees of the BANDED SQRT strategy.

**Proposition E.2.** *Let $\mathbf{C}(\boldsymbol{\theta}_0)$ denote the strategy for BANDED SQRT and note that it is a banded Toeplitz matrix. Now let $\mathbf{C}(\boldsymbol{\theta}_*)$ denote the banded Toeplitz matrix numerically optimized for the*

*objective function specified in Proposition 3.1, using $\boldsymbol{\theta}_0$ as the initialization. Then the expected error of $\mathbf{C}(\boldsymbol{\theta}_*)$ is never worse than the expected error of $\mathbf{C}(\boldsymbol{\theta}_0)$, that is:*

$$\|\mathbf{C}(\boldsymbol{\theta}_*)\|_{1,2}^2 \|\mathbf{AC}(\boldsymbol{\theta}_*)^{-1}\|_F^2 \leq \|\mathbf{C}(\boldsymbol{\theta}_0)\|_{1,2}^2 \|\mathbf{AC}(\boldsymbol{\theta}_0)^{-1}\|_F^2$$

*Proof.* First note that the objective we optimize is identical to the one shown in the proposition above, and note that it is a product of two continuous and differentiable functions of $\boldsymbol{\theta}$, and is therefore also continuous and differentiable. For simplicity, assume we optimize this objective with gradient descent. In each iteration, we compute the gradient which points in the direction of steepest descent. With sufficiently small step sizes, the sequence of iterates are guaranteed to be non-inceasing. Thus, the objective function of the final iterate is less than or equal to the objective function at the initial point. $\square$

Combining Proposition E.1 with Proposition E.2 shows that our banded Toeplitz strategy inherits the theoretical guarantees of the BANDED SQRT mechanism Kalinin & Lampert (2024). For simplicity, we presented these propositions for the Prefix workload, but Proposition E.1 can be generalized to handle any momentum-cooldown workload Kalinin & Lampert (2024), and Proposition E.2 applies for any workload.

# F ADDITIONAL EXPERIMENTS

## F.1 AMPLIFICATION VS NOISE CORRELATION

In Figure 6, we measure the benefit of privacy amplification and noise correlation respectively, by comparing the RMSE of Amplified DP-BANDMF with Unamplified DP-BANDMF and DP-SGD. Our main findings from this experiment are:

1. Amplification is most beneficial when both $\epsilon$ and epochs are small. However, even up to $\epsilon = 8$, there is a meaningful improvement in RMSE of $\approx 15\%$. Figure 6b shows that correlated noise is most beneficial when $\epsilon$ is large and epochs is small, and lines up very nicely with Figure 2b. In the most extreme setting, the RMSE of DP-BANDMF can be $9\times$ lower than DP-SGD. With larger epochs or smaller $\epsilon$, the benefit of correlated noise decreases, confirming it is most useful in compute-constrained settings.

2. When the number of epochs is large (but still well less than $n$), Amplified DP-SGD and Unamplified DP-BANDMF both achieve similar RMSE (as indicated by the dark blue region), despite being very different mechanisms. Moreover, the RMSE in these settings nearly matches the RMSE attained by full-batch DP-SGD, even with many fewer epochs (512 instead of 16384).

## F.2 COMPARISONS TO BASELINES

In Figure 1b of the main text, we compared our scalable amplified DP-BANDMF approach with other strategies as a function of $\epsilon$. In this section, we compare against the same baselines, but vary the number of training iterations $n$ and the number of epochs $k$.

In Figure 7a, we see that the DP-BANDMF improves relative to baselines as $n$ increases. At $n = 10^5$, our mechanism enjoys 25% smaller RMSE than the next best competitor. In Figure 7b we see that DP-BANDMF enjoys the greatest relative improvement over competitors for small $k$, although the relationship is not monotonic.

## F.3 BANDS VS. EPOCHS VS. RMSE

In this section, we revisit Figure 2a but increase $\epsilon$ from 1 to 8. Figure 8 shows how the number of bands effects the RMSE under different number numbers of epochs. With larger $\epsilon$, more epochs are needed for DP-SGD to get near full-batch performance (in this case, 256 epochs). The optimal number of bands also tends to be somewhat higher in this regime, with 128 bands being optimal for

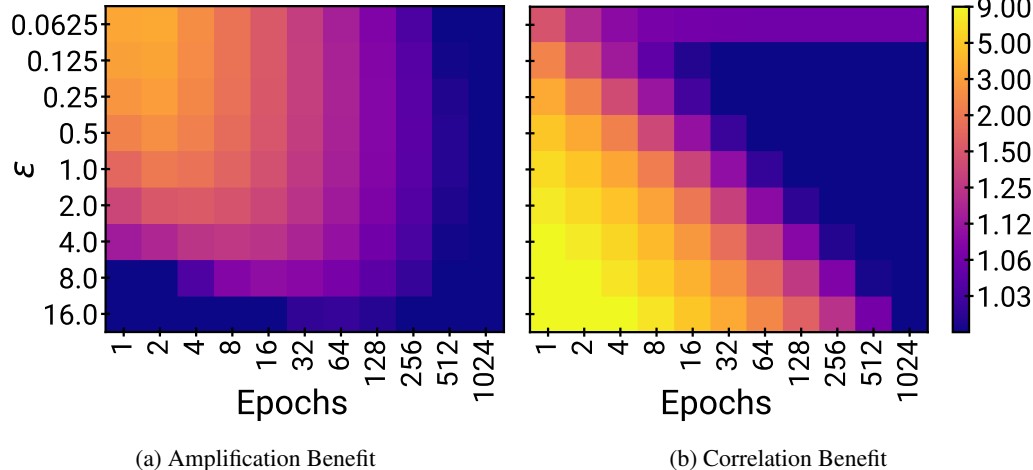

(a) Amplification Benefit        (b) Correlation Benefit

Figure 6: (a) Ratio of RMSE between Unamplified DP-BANDMF ($b = n/k$) and DP-BANDMF, which measures the benefit of amplification in different settings. (b) Ratio of RMSE between Amplified DP-SGD ($b = 1$) and DP-BANDMF, which measures the benefit of correlated noise in different settings. Both plots show results for $n = 16384$.

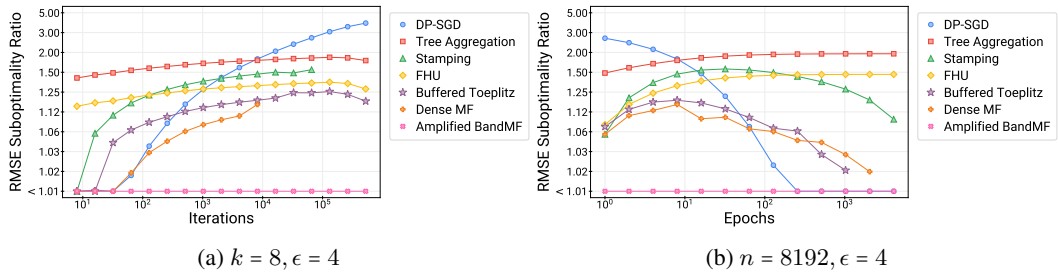

(a) $k = 8, \epsilon = 4$        (b) $n = 8192, \epsilon = 4$

Figure 7: RMSE Suboptimality ratio of different strategies as a function of $n$ (left) and $k$ (right) for $\epsilon = 4$.

the 8 epoch setting. DP-BANDMF needs 16 epochs to get within a factor of $2\times$ of the RMES of full-batch DP-SGD.

## F.4 MAX ERROR VS. LEARNING PERFORMANCE

## G QUALATATIVE COMPARISON TO PRIOR AND CONCURRENT WORK

There are three primary axes we can evaluate prior matrix factorization approaches on: (1) efficiency of strategy selection, (2) training-time overhead, and (3) utility. Table 2

The factorizations that are most efficient to calculate avoid numerical optimization Abadi et al. (2016); Kairouz et al. (2021b); Kalinin & Lampert (2024); Fichtenberger et al. (2023), but do so at the cost of utility. The factorizations that are most expensive to compute represent the noise correlation strategy using dense matrices, and are generally inefficient for large $n$ Denisov et al. (2022); Choquette-Choo et al. (2023b; 2024). Our proposed approach, as well as one concurrent approach McMahan et al. (2024a;b) get the best of both worlds, by numerically optimizing over parameterized classes of strategies that avoid dense representations.

Different factorizations have different properties that can be exploited for runtime efficiency. DP-SGD is the most efficient method, since it adds independent rather than correlated noise. The most general matrix factorizations require time linear in $n$ to generate noise for a *single iteration*, which can be prohibitively expensive for large $n$ (even with distributed noise generation). Indeed, these

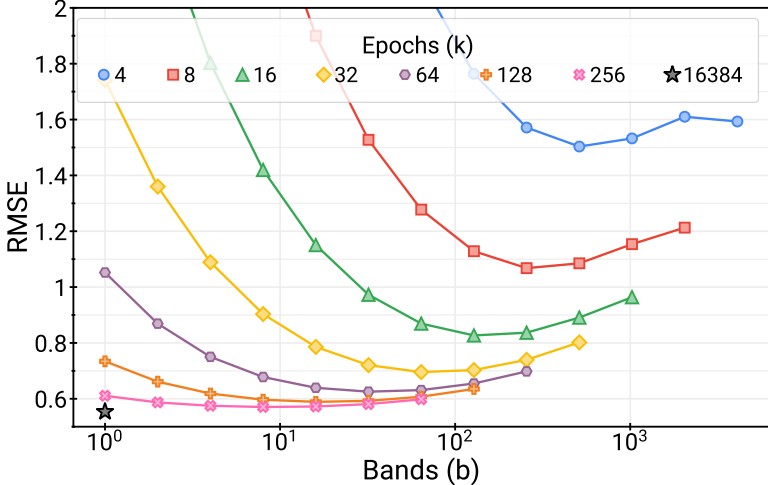

Figure 8: RMSE of DP-BANDMF as a function of $b$ for various epochs, with fixed $(\epsilon, \delta) = (8, 10^{-8})$ and $n = 16384$. With $b = 1$, DP-BANDMF is equivalent to DP-SGD, and only benefits from amplication (not correlated noise). With $b = \frac{n}{k}$, DP-BANDMF only benefits from correlated noise (not amplification) and closely resembles DENSE-DP-MF. The best RMSE is obtained somewhere in the middle, with the both the best RMSE and corresponding value of $b$ decreasing with epochs. By using DP-BANDMF instead of DP-SGD, one can run for far fewer epochs without sacrificing nearly as much utility.

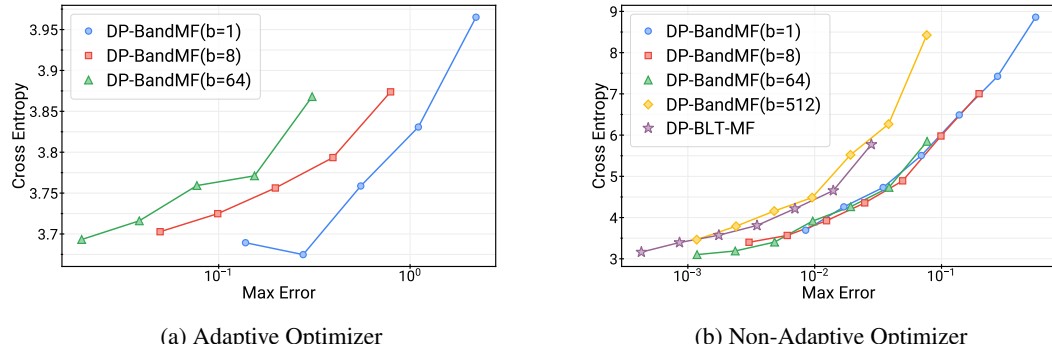

(a) Adaptive Optimizer

(b) Non-Adaptive Optimizer

Figure 9: Max Per-Query Error vs. Learning Performance (evaluation cross entropy) with an adaptive optimizer (a) and a non-adaptive optimizer (b). In both (a) and (b), a 4M parameter `BertTiny` model is trained on the StackOverflow dataset for various noise multipliers.

prior works focused on settings where $n \leq 2052$ Denisov et al. (2022); Choquette-Choo et al. (2023b). Our method bypasses this limitation via Algorithm 1, and enjoys only a $b\times$ overhead.

Finally, different factorizations enjoy different utility. We say a factorization is optimal in at least one setting of ($n \leq 2^{20}$, $k \leq n$, $\epsilon \leq 16$) if it achieves the lowest RMSE among all methods that scale to that setting. While this characterization doesn't quantify how close to optimal different methods are, it clearly demonstrates the advantages of DP-BANDMF relative to other scalable approaches.

| Factorization | Efficient Strategy Selection | Sub-linear Training Overhead | Optimal in at least one setting |
|---|:---:|:---:|:---:|
| Identity (DP-SGD) | ✓ | ✓ | ✗ |
| TREE AGGREGATION Kairouz et al. (2021b) | ✓ | ✓ | ✗ |
| FHU Fichtenberger et al. (2023) | ✓ | ✗ | ✗ |
| LTI Choquette-Choo et al. (2023a) | ✓ | ✗ | ✗ |
| STAMPING Denisov et al. (2022) | ✗ | ✗ | ✗ |
| MULTI-EPOCH MF Choquette-Choo et al. (2023b) | ✗ | ✗ | ✓ |
| BLT-DP-MF McMahan et al. (2024a) | ✓ | ✓ | ✗ |
| BANDED SQRT Kalinin & Lampert (2024) | ✓ | ✓ | ✗ |
| **DP-BANDMF [Ours]** | ✓ | ✓ | ✓ |

Table 2: Summary of existing matrix factorizations and their performance characteristics.

## H  IMPROVING FRAGILITY TO MIN SEPARATION FOR FEDERATED TRAINING SCENARIOS

The primary focus of this paper has been on centralized training regimes, where DP-BANDMF benefits from privacy amplification by sampling. As observed in prior work (Choquette-Choo et al., 2024; McMahan et al., 2024b) and in Figure 1b Unamplified DP-BANDMF is still a state-of-the-art mechanism in federated training regimes. There, a $(k, b)$-min-separation participation pattern is used where each user contributes at most $k$ times during training, and each contribution is separated by at least $b$ iterations. In centralized scenarios we know what these parameters are in advance based on properties of the dataset and training setup. In federated scenarios, it is more difficult to control, and therefore the strategy is typically optimized based on estimates of these quantities, while the privacy budget consumed during training is computed post-hoc (Xu et al., 2023). If the min separation is correctly estimated, DP-BANDMF is the best known mechanism currently available. However, if the true min separation differs from the one the strategy was optimized for significantly (e.g., more than a factor of 2×), BLT-DP-MF can outperform DP-BANDMF.

Fundamentally, this phenomenon is due to the fact that the Toeplitz coefficients in the BLT strategies decay more quickly than the ones in banded Toeplitz strategies. A simple heuristic can improve the robustness of banded Toeplitz strategies to miscalibration of min-separation, at the cost of increased expected error when the min-separation is correctly calibrated.

- Let $b_0$ and $b$ denote the lower and upper bound on min separation we want to tailor the mechanism for.
- Optimize $\boldsymbol{\theta} \in \mathbb{R}^b$ for expected error as in Proposition 3.1.
- Set $\theta_i = \frac{b-i}{b-b_0}\theta_i$ for all $i \geq b_0$.

We take the setting considered in McMahan et al. (2024b), where $n = 2000$, $k = 5$, and min-sep is between $b_0 = 200$ and $b = 400$. In Figure 10a we plot the Toeplitz coefficients of three strategies: BLT, BandToep optimized for $b = 400$, and BandToep + Heuristic described above. In Figure 10b we plot the RMSE of each strategy as a function of the minimum separation. With our heuristic, we achieve lower RMSE than BLT strategies across all min separations considered, but sacrifice some RMSE over the baseline banded Toeplitz strategy when the minimum separation is very close to 400. Note the relative differences between all three mechanisms are pretty small in this setting. Finally, we will note that the heuristic can likely be replaced with something more principled, by e.g., optimizing for the expected error averaged over different min-separations. We leave this exploration for future work.

## I  SCALING BEYOND $n > 10^7$

Beyond $n > 10^7$, numerically optimizing banded Toeplitz strategies becomes expensive using the techniques we described in the main body. With a small observation, we can derive an approximate loss function whose time complexity is significantly reduced. Recall from Proposition 3.1 that to

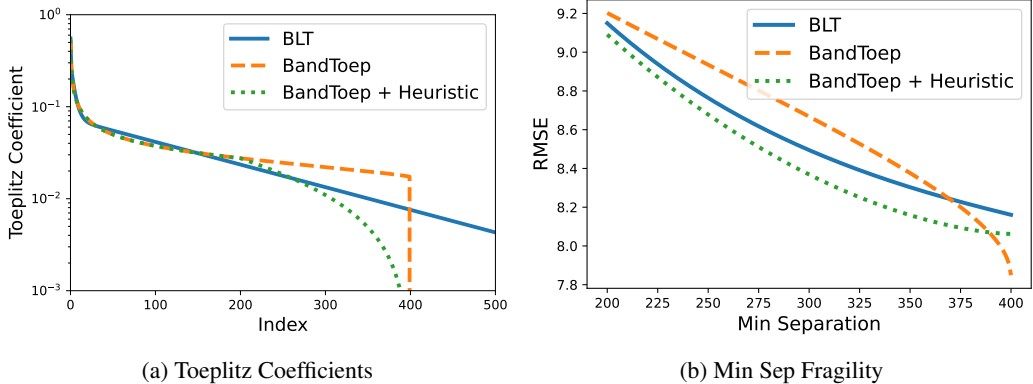

(a) Toeplitz Coefficients

(b) Min Sep Fragility

Figure 10: Comparison between Buffered Toeplitz strategies and Banded Toeplitz strategies when the min-separation is not known in advance (i.e., federated training scenarios).

compute the expected error of a Toeplitz strategy we need to compute $\mathbf{y} = \mathbf{C}(\boldsymbol{\theta})^{-1}\mathbf{1}$ (if we are optimizing for the Prefix workload). Note that $\mathbf{y}_i$ is defined by a simple linear recurrence of order $b$ shown in Algorithm 1. Rewriting that recurrence here, we have:

$$y_i = (1 - \sum_{j=2}^{b} \theta_j y_{i-j+1})/\theta_1$$

Now consider the setting where $n \gg b$. Now suppose for a moment the sequence $y_i$ converges to some fixed point $x$, that is $\lim_{i\to\infty} y_i = x$. We can solve for this fixed point by substituting $y_i = y_{i-j+1} = x$, and solving for $x$. Doing so, we obtain:

$$x = \frac{1}{\sum_{j=1}^{b} \theta_j}$$

It is natural to question whether the sequence $y_i$ converges. Since divergence would generally imply high expected error, we believe any reasonable iterative optimization procedure should produce well-behaved strategy parameters $\boldsymbol{\theta}$. To compute an approximate expected error, we can simply compute $y_i$ exactly up to some fixed index $m$, and then approximate $y_i \approx \frac{1}{\sum_j \theta_j}$ for $i > m$. Using this idea, we can easily approximate the expected mean squared error as:

$$\frac{1}{n}\sum_{i=1}^{n} i y_i^2 \approx \frac{1}{n}\Big[\sum_{i=1}^{m} i y_i^2 + \Big(\frac{1}{\sum_j \theta_j}\Big)^2 \sum_{i=m+1}^{n} i\Big]$$

We can evaluate this expression in $O(m)$ time using the closed form expression for $\sum_{i=m+1}^{n} i$. We can similarly easily approximate the expected max squared error as:

$$\sum_{i=1}^{n} y_i^2 \approx \sum_{i=1}^{m} y_i^2 + \Big(\frac{1}{\sum_j \theta_j}\Big)^2 (n - m + 1)$$

Empirically, the sequence converges quite quickly and we recommend setting $m = b^2$ (assuming $n > b^2$), which gives an overall time complexity of $O(b^3)$ for evaluating the loss function. As a concrete data point, optimizing the expected max squared error for $n = 2^{16}$ and $b = 64$ using $m = b$, $m = b^2$ yields suboptimality ratios are 1.032 and 1.0 respectively. That is, setting $m = b^2$ gives no loss in solution quality.

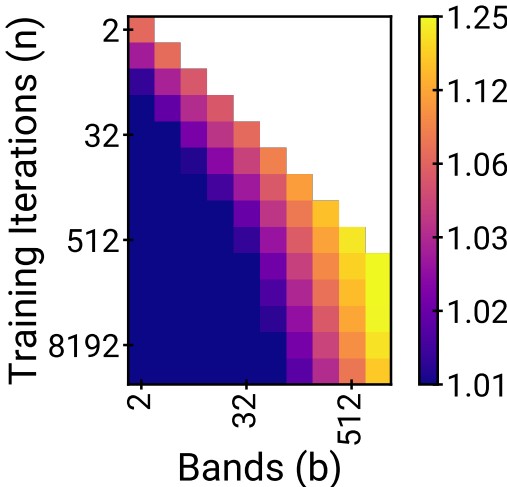

Figure 11: RMSE Suboptimaltiy Ratio of the optimal parameterized banded Toeplitz strategy of the form $\theta_i = 1/i + c$ , and the optimal unconstrained banded Toeplitz strategy, for various choices of $n$ and $b$.

## J  NEAR-OPTIMAL BANDED TOEPLITZ STRATEGIES WITHOUT GRADIENTS

Through inspection of the optimal Toeplitz coefficients in Figure 10a, one can attempt to fit a functional form to the parameters. The expression $\theta_i = 1/i + c$ provides an excellent fit for the optimal banded Toeplitz coefficients when $c$ is chosen appropriately. To find the best value of $c$ given the $n$ and $b$, one can do a simple sweep over a range of possible values, picking the best one. This avoids the need to compute gradients, and the evaluation of the loss function at different values of $c$ can be easily vectorized / parallelized. This provides an excellent approximation of the optimal Toeplitz coefficients, particularly when $n >> b$, as demonstrated in Figure 11.

While we do not work out the details here, we believe that it may be possible to identify an efficient single-shot algorithm to find the optimal value of $c$ by deriving and analyzing the gradient, and identifying where it is $0$.

## K  OPTIMIZING COLUMN-NORMALIZED BANDED TOEPLITZ STRATEGIES

In Section 3.2, we discuss the importance of column normalization, and propose a simple heuristic of normalizing the columns of the optimized banded Toeplitz matrix. This heuristic improves the expected error over the corresponding un-normalized strategy, but is not in general optimal among the class of all column-normalized banded Toeplitz strategies, (since it was optimized without accounting for column normalization). In this section, we describe two approaches to optimize over the space of column normalized banded Toeplitz strategies. In general, we *can* optimize over the space of column-normalized Toeplitz strategies, but we would have to forfeit the efficiency advantages that come along with the Toeplitz structure, and instead use the techniques we described for general banded matrices in Section 3.1. Alternatively, we can use the gradient-free approach described in App. J, which can be helpful because the gradient computation is the primary bottleneck for general banded optimization as explained in App. C. However, the latter approach would only give an approximate minimum (although in practice the sub-optimality is likely too small to matter).

