# OpenReview forum: "Scaling up the Banded Matrix Factorization Mechanism for Large Scale Differentially Private ML"
_ICLR.cc/2025/Conference — ICLR 2025 Spotlight_

### Official Review · Reviewer_YuCq · 2024-11-03

**Soundness:** 3
**Presentation:** 3
**Contribution:** 3
**Rating:** 8
**Confidence:** 2

**Summary:**

This paper proposes a differential privacy method that utilizes both random sampling and correlated noise via the use of b-banded strategy matrix. The number of bands b controls the proportion of privacy amplification from subsampling and correlated noise, which can be optimally selected with efficient computation cost using the banded Toeplitz strategy. Further distributed noise generation is used to save potential memory cost.

**Strengths:**

1. The problem is well explained and motivated.
2. Extensive theoretical and empirical analysis to support the proposed mechanism.
3. The paper is well-written and easy to follow.

**Weaknesses:**

1. A discussion on the communication cost w.r.t the number of bands as a tradeoff in the distributed setting would be nice to have.

**Questions:**

I have no questions.

---

> ### Author Response · Authors · 2024-11-18
>
> Thank you for the feedback, we are glad you liked the paper.  We will be sure to incorporate a discussion of the communication cost of our distributed noise generation procedure in Section 3.3.

---

> > ### Comment · Reviewer_YuCq · 2024-11-26
> >
> > Thank you, I will maintain my score.

---

### Official Review · Reviewer_rykK · 2024-11-04

**Soundness:** 3
**Presentation:** 3
**Contribution:** 3
**Rating:** 6
**Confidence:** 2

**Summary:**

The paper presents a improvement to the DP-BANDMF, a differentially private mechanism that adds correlated noise to DP-SGD, aiming to address its scalability limitations. Existing approach DP-BANDMF has struggled with computational and memory demands, especially in large-scale models. The authors introduce two methods  to optimize this mechanism for scenarios involving over 10^6 training iterations and up to 10^9 model parameters, making it feasible for use with modern, large-scale models. The empirical results demonstrate significant performance gains over existing mechanisms.

**Strengths:**

1. This paper is well written and clearly addresses the contributions.
2. The empirical study is thorough with limitations sufficiently addressed.

**Weaknesses:**

The only concern here is that this paper does not discuss too much privacy utility trade-off, which is not the focus of this paper.

**Questions:**

1. Is there any insight on why adaptive estimator works worse than adaptive optimizer?

2. In practice, how do we manage the privacy budget for selecting number of bands?

---

> ### Author Response · Authors · 2024-11-18
>
> Thank you for taking the time to review this paper.  Below we respond to the weaknesses and questions raised:
>
> 1. The broader approach considered in this paper of DP-MF is motivated by improving privacy/utility/compute trade-offs in private machine learning applications.  We will be sure to emphasize this more in our introduction.
>
> 2. We do not fully understand your first question, if you can clarify we’d be happy to discuss further.
>
> 3. One nice thing about our approach and DP-BandMF more broadly is that we can select the optimal number of bands without consuming any privacy budget by minimizing the RMSE, which is a data-independent proxy for learning performance.

---

> > ### Comment · Reviewer_rykK · 2024-11-24
> >
> > Thank you for your clarification! Apologies for the typo in my question—I meant to ask why the adaptive optimizer performs worse than the non-adaptive optimizer in Figure 3. After re-reading the paper, I see that Section 6 answers this well, noting that RMSE is not always a reliable proxy for learning performance with adaptive optimizers. Thanks again for the detailed response!

---

### Official Review · Reviewer_HJgB · 2024-11-06

**Soundness:** 3
**Presentation:** 4
**Contribution:** 3
**Rating:** 8
**Confidence:** 3

**Summary:**

The paper studies a mechanism (DP-BandMF) for private machine learning that has advantages over the standard private mechanism (DP-SGD) in some regimes due to its use of optimized correlated noise. The algorithm is characterized by a strategy matrix that determines the correlational structure of the noise.

This work identifies the optimization of the stategy matrix is a computational bottleneck limiting the applicability of DP-BandMF. Prior work gives an $O(n^3)$ time and $O(n^2)$ space algorithm, which is impractical for large values of $n$. This work improves the running time to $O(bn^2)$ and the space to $O(bn)$ where the band size $b$ characterizes the level of correlation allowed between noise vectors. The authors go on to give a further improved $O(bn)$ time $O(n)$ space algorithm for a restricted class of strategies.

The authors conclude with a series of experiments that assess the scalability and solution quality of their algorithm, the optimal band-size, as well as the suitability of the RMSE measure optimized by their algorithm as a proxy for utility loss.

**Strengths:**

The paper investigates practical scalability issues of a useful DP-ML algorithm and makes substantial performance improvements that increase the range of high-dimensional learning tasks that may be solved by DP-BandML.

The purpose and conclusions of the experiments are well-explained.

Overall, the paper is very clearly written and pleasant to read.

**Weaknesses:**

A small point not addressed in this work is efficient computation of the gradient of the RMSE objective. The authors defer to the Jax implementation. It is unclear whether there is an inherent limitation of this approach or if there is room for meaningful improvement in gradient computation efficiency.

A more significant weakness of this work is somewhat limited technical novelty in the results. The primary technical contribution appears to be Algorithm 3, which leverages sparsity and computes the objective in a streaming fashion.

The authors do extend their results in Proposition 3.1 to a new setting involving Toeplitz strategies. This result is nice but I found the following motivation not fully convincing: "This design decision was inspired by manual inspection of the optimal dense strategies, observing that they exhibit a near-Toeplitz structure." While this choice seems bolstered by by the result in Figure 1(a), a more careful theoretical justification would be welcome, if possible.

**Questions:**

- Could context be provided for how a "strategy" should be interpreted? Around l85 in the background.
- The "workload" $A$ is introduced around l150 but the context is also unclear to me here. What is the role of this object and why is it natural to view as a lower triangular matrix of ones?
- Could the authors provide a definition of Toeplitz strategies? One was not provided.
- Lastly, is there a typo on l85? $i \leq j + b$ looks the wrong-way-around to me.

---

> ### Author Response · Authors · 2024-11-18
>
> Thank you for your thoughtful review and detailed feedback, you raise a number of good points which we discuss further below.
>
> 1. You are correct to point out that we did not really elaborate on the gradient computation of our MSE objective, and that directly using Jax’s reverse-mode autodifferentiation capabilities may lead to a suboptimal computation of gradients.  For non-toeplitz strategy optimization, we actually had to be pretty careful with our implementation to get around one such inefficiency.
> In particular, while Algorithm 3 requires only O(b n) memory, by default when backpropagating through this function to compute gradients Jax keeps around all intermediate iterates, and hence uses O(b n^2) memory.  We got around this issue by using a technique called “checkpointing” which trades off time for memory during back-propagation through for loops.  We configured the number of checkpoints to ensure that the memory consumed during the backwards pass never surpassed 4GB.
> This subtle implementation trick was not needed for banded Toeplitz optimization; there we did just use Jax’s gradients directly.  We thank the reviewer for pointing this out, these implementation details are important for reproducibility and hence we will write a paragraph in the appendix on this topic, and add a forward pointer to it from Section 3 where we discuss gradients.
>
> 2. The reviewer’s critique on the technical novelty is certainly valid, and we agree with the reviewer that our core approach is pretty simple, although we do view that as a strength rather than a weakness.  We are glad that you found the strengths of our work to outweigh this limitation.
>
> 3. In Section D, we do provide some theoretical guarantees that may be of interest to you.  We will add a forward pointer to them as a footnote on the informal justification for the design decision.
>
> Thank you for the additional questions/feedback, we will be sure to incorporate these points into our revised paper.

---

### Meta-Review · Area_Chair_JFKZ · 2024-12-16

**Metareview:**

The paper proposes techniques to allow scaling up matrix mechanisms for differential privacy to significantly larger problems than previously.

This is potentially a very valuable contribution, as matrix mechanisms provide superior privacy-utility-tradeoff compared to standard DP-SGD, and the paper addresses one of their major weaknesses in computational cost.

The reviewers do not identify any significant weaknesses in the paper, and it should clearly be accepted.

**Additional Comments On Reviewer Discussion:**

There was essentially no discussion as all reviewers recommended acceptance.

---

### Decision · Program_Chairs · 2025-01-22

Accept (Spotlight)